# Enhancing Operational Efficiency and Service Delivery through a Robotic Dispensing System: A Case Study from a Retail Pharmacy in Brazil

**DOI:** 10.3390/pharmacy12050130

**Published:** 2024-08-23

**Authors:** Karen Basile, Monserrat Martínez, Julia D. Lucaci, Claudia Goldblatt, Idal Beer

**Affiliations:** 1Medical Affairs MMS, Becton Dickinson, Sao Paulo 04717, SP, Brazil; karen.basile@bd.com; 2Medical Affairs MMS, Becton Dickinson, Ciudad de Mexico 11000, Mexico; 3Health Economics and Outcomes Research, Becton Dickinson, Franklin Lakes, NJ 07417, USA; 4Medical & Scientific Affairs, Becton Dickinson, San Diego, CA 92121, USA

**Keywords:** pharmacy automation, dispensing robot, robotic dispensing system, automated dispensing system, pharmacy workflow, automation

## Abstract

Drug dispensing in retail pharmacies typically involves several manual tasks that often lead to inefficiencies and errors. This is the first published quality improvement study in Latin America, specifically in Brazil, investigating the operational impacts of implementing a robotic dispensing system in a retail pharmacy. Through observational techniques, we measured the time required for the following pharmacy workflows before and after implementing the robotic dispensing system: customer service, receiving stock, stocking inventory, separation, invoicing, and packaging of online orders for delivery. Time savings were observed across all workflows within the pharmacy, notably in receiving stock and online order separation, which experienced 70% and 75% reductions in total time, respectively. Furthermore, customer service, stocking, invoicing, and packaging of online orders, also saw total time reductions from 36% to 53% after implementation of the robotic dispensing system. This study demonstrates an improvement in the pharmacy’s operational efficiency post-implementation of the robotic dispensing system. These findings highlight the potential for such automated systems to streamline pharmacy operations, improve staff time efficiency, and enhance service delivery.

## 1. Introduction

In most retail pharmacies, the drug dispensing process includes verifying the appropriateness of the prescription, manually selecting medicines from shelves, collecting payment, and packaging the medication. Additionally, pharmacists or technicians are often responsible for managing stock [1]. This workflow management includes inventory control (e.g., management of expired medications), purchasing, receiving stock, stocking inventory and its cleaning, and controlling drug dispensing. Due to the multitude of tasks being tended to by pharmacy staff, errors and inefficiencies can occur at any point [2,3].

The integration of automation techniques for medication management processes, such as ordering, dispensing, delivering, and administering medications, has emerged as an important strategy to improve the overall functionality and operations of inpatient, outpatient and retail pharmacies [4,5]. This approach not only improves operational aspects but also allows pharmacists to focus more on value-added activities, which can include providing detailed counseling on medication usage, discussing potential side effects, and offering tailored counseling about effective medication management [6,7]. Moreover, automating routine tasks substantially reduces patient waiting times, thereby enabling pharmacies to serve more customers without compromising service quality [5]. Optimizing pharmacy workflows to improve efficiency may result in increased patient satisfaction, as it enables clients to receive prompt service and personalized care [8]. This increase in satisfaction is crucial, as it contributes to higher patient loyalty and improves medication adherence [9].

Despite the benefits mentioned above, and the widespread availability of these devices across the world, the effectiveness of automated dispensing systems, particularly in retail settings, requires further evaluation. It is important to understand the advantages of implementing automated dispensing systems to appropriately assess their impact on patient care and pharmacy management.

This article aims to specifically analyze the efficiency enhancements in operational parameters reflecting core areas of service delivery and stock management introduced by a robotic dispensing system in a retail pharmacy in Brazil.

## 2. Materials and Methods

### 2.1. Study Design

This prospective, single site, pre- and post-observational quality improvement (QI) study was designed to assess the impact of a robotic dispensing system on pharmacy workflows and the process times associated with each. Using observational techniques, the time required to perform relevant pharmacy activities was measured onsite by an external observer (from a third-party vendor not affiliated with the study site or study sponsor) before and after the installation of a robotic dispensing system. During both the pre- and post-implementation phases, the pharmacy staff consisted of a pharmacist, a technician, and three employees dedicated to inventory and stocking all performing activities that are expected in one’s job description. The pre-implementation visits took place in November 2021, and post-implementation visits occurred in August 2022. Each visit measured one specific pharmacy workflow. No changes to standards of care were implemented. No customers were involved in responding to questions or asked to perform any different tasks as part of this project. No customer or staff member information was collected or could be identified, directly or through identifiers.

### 2.2. Setting

The study was conducted in a privately owned, independent medium size retail pharmacy in São Paulo, Brazil. In addition to serving customers in-person and online, the pharmacy functions as a distribution center for four subsidiaries. This retail pharmacy sells pre-packaged medicines that do not need compounding, and it also offers a variety of other items, including personal hygiene, perfume, cosmetics and dermocosmetics, to customers. Online orders are received through an electronic system and are delivered to the customer’s house via a delivery service. 

Prior to the intervention, a pharmacist or qualified technician was responsible for verifying the prescription and entering it into an in-house computerized prescriber order entry system (Linx, São Paulo, Brazil) when it was received. The pharmacy staff then manually checked the availability of the prescribed medication within the pharmacy’s inventory and collected the ordered items. A barcode-controlled system integrated to a specialized pharmacy management system (Linx, São Paulo, Brazil) enabled technicians to prevent errors during manual dispensing. Once prescriptions were filled, staff members delivered them to customers at the counter. In addition, the pharmacy staff manually managed storage, inventory, and stock cleaning (of the shelves and of each medication box on a quarterly basis) using more traditional methods. The same barcode-controlled system was used to scan and prepare online orders and to process supplier invoice receipts.

After the robotic system was installed, all packages, aside from controlled substances and refrigerated medications, were dispensed using barcode recognition. Pharmacy staff continued receiving and verifying the appropriateness of the prescription and delivering filled prescriptions to customers waiting at the counter. Medication packages delivered to the facility were automatically stored after being measured and having their barcodes scanned. Updated stock reports were routinely generated by the robot for pharmacists based on inventory and pre-determined minimum stock levels to prevent potential product stock-outs. Moreover, the presence of a cleaning module enabled the robot to automatically clean medication boxes and shelves, eliminating the need for pharmacy staff to engage in this activity.

### 2.3. The Robotic Dispensing System

The intervention consisted of the installation of a robotic dispensing system, including a storage and dispensing system (BD Rowa™ Vmax 160, BD Rowa, Kelberg, Germany), conveyor system (BD Rowa™ Conveyor System, BD Rowa, Kelberg, Germany), and a second input belt at the pharmacy (Figure 1).

The robotic dispensing system is an automated storage and retrieval system that can be customized to fit various pharmacy dimensions and layouts. The system can process both rectangular and round pack sizes, has a stock input of approximately 3 s, and output times between 8 to 12 s, enabling the simultaneous dispensing of up to nine packs. The second picking head of the robot can also allow simultaneous preparation of multiple orders, further enhancing the throughput of the pharmacy’s dispensing process. There are more than 13,000 similar systems implemented across the globe. 

The conveyor system, including a servo-driven lift mechanism and a spiral chute with a diameter of 400 mm, facilitated the swift transport of medications from the robotic system to the point of sale or dispensing area.

### 2.4. Pharmacy Workflow Endpoints

To evaluate the efficiency enhancements introduced by the pharmacy automation through a robotic dispensing system within the retail pharmacy setting, we performed a pre- and post-implementation time analysis of operational parameters reflecting the core areas of service delivery and stock management during the times of operation of the pharmacy. The pre-implementation measures were obtained in November 2021, before the robot installation that occurred in January 2022 and the post-implementation measures were made in October 2022. A time study technique was applied where one external observer measured the duration of each task with a stopwatch focusing on the following pharmacy workflows and measurements (see Figure 2):The customer service workflow at the pharmacy counter was measured by tracking task durations from customer interaction initiation to medication dispensing.The receiving stock process was evaluated from supplier invoice receipt to system entry completion.The inventory stocking was assessed from the first medication package placement to the last.Separation of online orders was measured from order receipt to item readiness for dispatch in the basket.Invoicing of online orders was timed from basket pick-up to invoice printing completion.The packaging of online orders was evaluated from item pick-up for packaging to order readiness for shipping.

### 2.5. Data Analysis

Descriptive statistics and 95% confidence intervals were calculated for all pharmacy workflows before and after the intervention using SAS version 9.4 (SAS Institute Inc., Cary, NC, USA). Confidence intervals were used to show the range of effect sizes supported by the observations and assess operational workflow significance. Graphical displays were created in Microsoft Excel (Microsoft, Redmond, WA, USA).

## 3. Results

This analysis comprised 111 time measurements across all evaluated pharmacy workflows. These measurements amounted to an aggregate measurement time of 22 h, 25 min, and 51 s.

Figure 3 displays the total time of all recorded measurements in each pharmacy workflow before and after the installation of the robotic system. A decrease in the total process time was observed across all workflows, but the most notable reductions were noted for activities related to receiving stock (−70%) and separation of online orders (−75%).

### Average Time Savings by Pharmacy Workflow after the Robotic Dispensing System Implementation

Figure 4 illustrates the average process time for each workflow before and after the implementation of the robotic system.

The difference in average measured time per workflow (min:ss) between the pre- and post-implementation periods, along with their 95% confidence intervals, are displayed above the brackets for each pharmacy workflow.

Average time savings (reported as min:ss) were observed across all pharmacy workflows. The most significant time savings were observed in the process times tied to receiving stock, as the installation of the system reduced the average time by 06:07 (from 08:44 pre-implementation to 02:37 post-implementation). Separation of online order processes also experienced an efficiency gain, with the time required decreasing by an average of 04:18—from 05:45 pre-implementation to 01:27 post-implementation. Additionally, stocking tasks showed enhanced efficiency due to the robotic system, evidenced by a reduced average time of 03:41, from 07:37 before implementation to 03:56 after implementation.

Post-implementation, the customer service workflow experienced a reduction of 00:34 on average. The robotic system also made the invoicing process for online orders more efficient, with an average time saving of 01:11. Lastly, the packaging of online orders workflow exhibited a decrease of 00:54 on average.

## 4. Discussion

This is the first study examining the impact of a robotic dispensing system in Latin America. This study focused on the operational efficiency of a retail pharmacy in Brazil—a lower-middle-income country characterized by diverse pharmaceutical care and early stages of pharmacy automation. We showed reductions in both total and average times for all evaluated retail pharmacy workflows, with the most significant time savings observed in workflows tied to receiving stock, inventory stocking and separation of online orders.

The reduction in time required for stock management processes and order separation was primarily due to the robotic system’s ability to bypass the need for manual shelf searching. With its advanced mapping, the robot precisely knows where each medication is stored, facilitating direct and efficient stocking. Furthermore, the system’s organization and space utilization led to quick medication access and eliminated the time staff would spend searching for products. Time savings are also achieved through the robot’s capability to stock multiple products simultaneously. Equipped with two input belts, the system allows for one belt to be loading while the other is stocking. This streamlines operations. The improved inventory accuracy through precise tracking and dispensing capabilities minimizes the time spent on manual inventory checks and corrections, allowing for optimal stock levels and reduced instances of overstocking or understocking. Furthermore, the robotic system’s capability to automatically monitor expiration dates and alert staff to imminent expiries promotes efficient stock rotation and eliminates the need for manual expiry checks.

Similar outcomes have been observed in previous studies examining the influence of robotic systems on pharmacy stock management. For example, Rodriguez-Gonzalez et al. [7] assessed the quality of stock management and staff satisfaction before and after the implementation of a robotic dispensing system in an outpatient hospital pharmacy in Spain. The study showed that post-implementation, the daily staff time spent on stock intake, storage, and order selection was reduced by 59.3%, from 01:36:15 to 00:39:10. Staff also reported high satisfaction with the robot introduction [7]. Bagattini and colleagues’ work [10] on the automation of a tertiary hospital pharmacy in Brazil highlighted similar advancements in operational efficiency by deploying automated systems. Implementing other automation technologies in the central pharmacy improved inventory management, reducing the frequency of breakages and losses of medicines and the number of expired products. Moreover, the robot deployment led to a decrease in overtime pay for the central pharmacy team.

In our study, process times related to invoicing and packaging of online orders also showed notable improvements post-implementation. The time reductions observed in these workflows can be attributed to the enhanced organizational layout of the invoicing and packaging environment post-implementation. Moreover, the robotic system reduced interruptions that previously occurred during these activities on a regular basis. These interruptions, also reported in previous studies [11,12], typically required staff to halt invoicing and packaging tasks to locate and deliver products to colleagues. It is important to note that while the total number of employees at the pharmacy remained the same during the pre- and post-implementation phases, there was a change in the employee responsible for packaging online orders. This change may have introduced a different work pace, contributing to the observed improvements in packaging time.

Upon examining the time savings across different pharmacy workflows after the implementation of the robotic dispensing system, customer service workflows exhibited the smallest time savings compared to the other measured activities. This is because customer service interactions involve direct communication and engagement with customers, which inherently require a human element. The reduction in time spent away from the counter to collect items, which the robotic system has positively impacted, is only one component of the overall customer service time. The other components that require personalized attention, advice, and instructions are not directly affected by the speed of medication retrieval. Thus, while pharmacy staffing remained constant after the implementation of the robotic dispensing system, the intervention may have increased the staff’s availability to engage with customers for additional periods of time that would have otherwise been spent completing manual activities. This shift towards longer interactions tailored to the specific needs of a customer may also have led to a positive effect on patient satisfaction and service quality [13,14,15,16]. Also important, an average 30 s reduction in time per customer interaction, while initially appearing small, can translate to a savings of one hour for every 120 customers served. This is especially beneficial during peak times in busy stores, where it can reduce waiting times and potentially enhance customer satisfaction [17]. Notably, patients who are satisfied usually adhere to the treatment prescribed, experience an enhanced quality of life, and continue using health services [9,18].

Regarding physical space reconfiguration, while not a specific endpoint for this study since it can vary depending on the size of the robotic system and square footage of the retail pharmacy, the implementation team estimated an initial space saving of approximately 107 square feet after the implementation at this retail pharmacy store that could potentially be repurposed.

While this quality improvement study provides valuable information into the efficiency gains from implementing a robotic dispensing system in a retail pharmacy, it has limitations inherent to its methodology and scope. For example, the study was conducted in a single retail pharmacy, limiting the diversity of operational environments, patient demographics, and pharmacy workflows considered. Additionally, while improvements were noted, it is important to acknowledge that not every enhancement observed in this study can be attributed directly to the robotic system. The pharmacy staff may have capitalized on the necessary changes to workflows and the spatial reconfiguration required to accommodate the robot, thereby leveraging the opportunity to review and refine workflows. Future research could benefit from including a larger, more varied sample of pharmacies to ensure thesefindings are representative of different settings. Subsequent studies should also examine the long-term impact of these systems on patient care and safety, customer and staff satisfaction, and the financial stability of pharmacy operations.

## 5. Conclusions

This quality improvement study indicates a consistent improvement in the pharmacy’s operational efficiency after the implementation of a robotic dispensing system. The adoption of this system resulted in less time spent on all retail pharmacy workflows, with the most significant improvements observed in receiving stock, separation of online orders, and stocking, where time savings reached up to 75%. Reductions in time were also seen in invoicing and packaging of online orders, contributing to improved productivity and customer service. These findings highlight the potential for such a system to streamline pharmacy operations, enable more efficient use of staff time, and enhance service delivery.

## Figures and Tables

**Figure 1 pharmacy-12-00130-f001:**
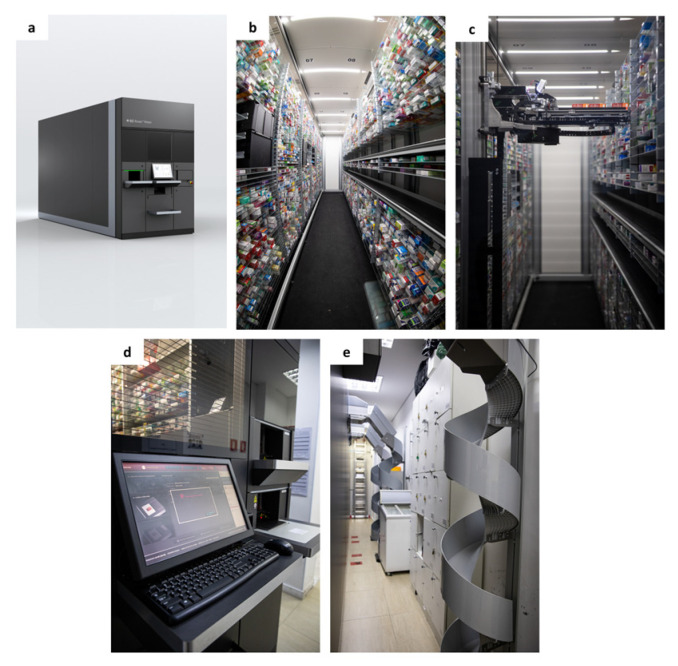
Robotic dispensing solution implemented. (**a**) Outside view of the automated storage and dispensing robot; (**b**) packages organized on shelves inside the robot; (**c**) picking head moving multiple packages; (**d**) touch-screen user interface and input belts; and (**e**) conveyor system with spiral chutes to transport packages from the upper floor to the ground floor.

**Figure 2 pharmacy-12-00130-f002:**
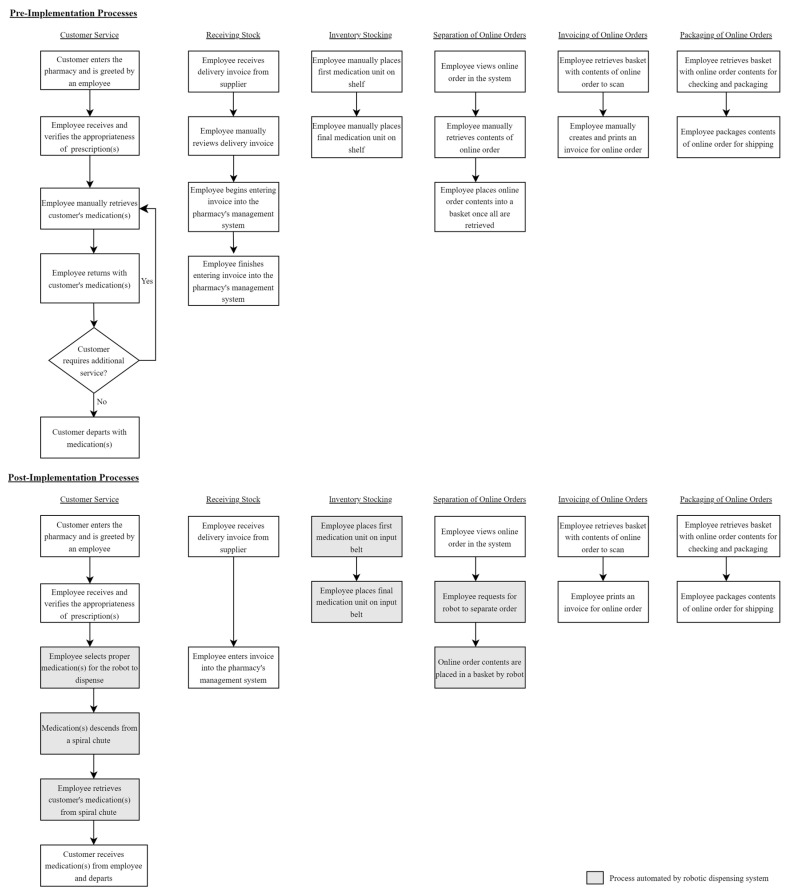
Pre- and post-implementation process maps for pharmacy workflows.

**Figure 3 pharmacy-12-00130-f003:**
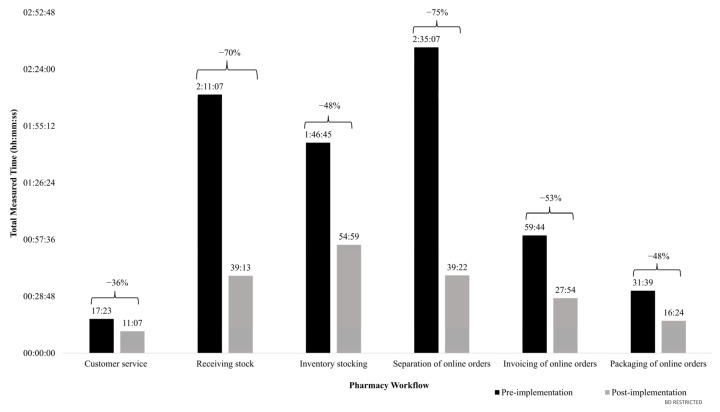
Total time savings by pharmacy workflow after installation of the robotic dispensing system.

**Figure 4 pharmacy-12-00130-f004:**
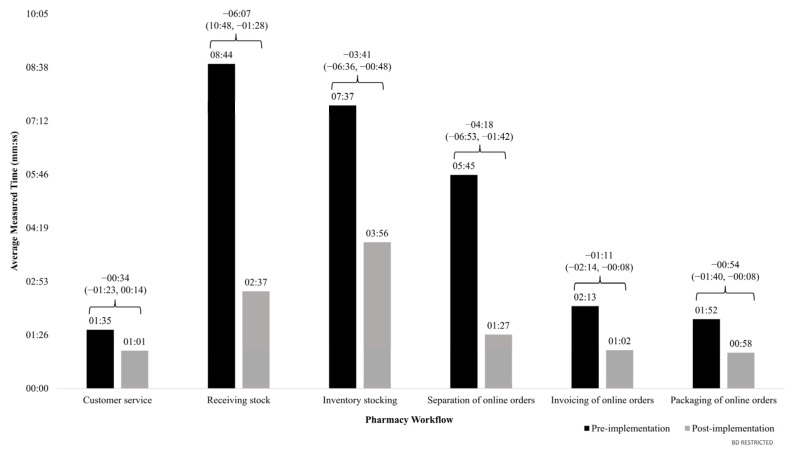
Average time savings by pharmacy workflow after the robotic dispensing system implementation.

## Data Availability

The data are available from the authors upon reasonable request.

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
