# Peer review of "Enhancing Operational Efficiency and Service Delivery through a Robotic Dispensing System: A Case Study from a Retail Pharmacy in Brazil"

_pharmacy, 2024, doi:10.3390/pharmacy12050130_

Round 1

Reviewer 1 Report

Comments and Suggestions for Authors

I appreciate the opportunity to read and review “Enhancing operational efficiency and service delivery through a robotic dispensing system: a case study from a retail pharmacy in Brazil”. This study evaluated the operational impacts of implementing a robotic dispensing system in a retail pharmacy in Brazil – the first study of it’s kind in Brazil. Investigators found improved operational efficiencies after implementation of the robotic system, a promising find for retail practice in Brazil. The comment below are intended to seek clarification in some areas of the paper, and offer further strength to this potential publication. Overall, a well-written paper.

1.     It may help to clarify somewhere in the manuscript the types of products being dispensed in the pharmacy, and subsequently by the robot. Based on the photos and as implied in the process, it appears that medicines are perhaps pre-packaged. This is a very different process from other countries that have robots that need to fill prescriptions by filling bottles from bulk stock (in other words, not pre-packaged). So any clarification that can be provided on the types of products being dispensed would be most helpful in understanding the workflow changes that came about from use of robotic technology.

2.     The authors note filling “online orders” several times throughout the manuscript. It may help to clarify what that might mean, or how that works for this particular, as different countries might use those terms differently. And, are there difference between online orders and other types of prescription orders? This could maybe be done around line 77.

3.     And to be clear, this study looked at online orders only, not in-person or distribution to the four subsidiaries mentioned in line 78?

4.    In line 76, it would be helpful if the authors could clarify what it means to be a medium size pharmacy in Brazil. Is it possible the authors could maybe provide what the average daily prescription volume (number of prescriptions) dispensed was? It would also be helpful if the authors could clarify whether this pharmacy is an independently owned pharmacy or part of a larger corporate structure.

5.    It would be helpful the authors talk a little more about how the data collection process actually occurred. Such as, was the time data collected by one investigator or more? Were data collectors aware of the study research question? Were they using a stopwatch or other equipment. Were the investigators documenting data into a coding sheet or computer? What dates did the data collection occur and time of day (may affect timing over rush hours, slow times, etc.)?

6.    The discussion and limitations section was well written!

Reviewer 2 Report

Comments and Suggestions for Authors

Thank you for the opportunity to review this manuscript. It is an interesting and valuable study and I enjoyed reading it. However, I have the following questions to be addressed before it is suitable for publication: 

Introduction: 

* You have provided a succinct overview of existing literature on the potential benefits of automated drug cupboards. However, it is not clear where this is from (hospital or community/outpatient/retail pharmacy and country) and therefore why you have chosen to focus on a retail pharmacy for your study. Please clarify this. 

* Please remove the overview of results from the last paragraph of your introduction. You can include more detail on your specific research questions or a more detailed rationale for your study here. 

Methods: 

* Please define what is meant by a medium sized retail pharmacy as this can vary by jurisdiction and is measured in different ways (e.g. script counts, staff number and mix or floorspace) 

* How long was each period of observation and when was this conducted? 

* Were any stastistical significance tests conducted? It is difficult to say that the robotic system led to significant improvements if no statistical tests were conducted. 

Other: 

* You have stated that an institutional review board (IRB) statement is not applicable. Please explain why IRB review is not needed here. 

* There are some minor spelling, typographical and grammatical errors in some parts (e.g. on line 106 of page 3 "customized to fit. various pharmacy dimensions..." should be "customized to fit various pharmacy dimensions..."). Please check for and correct these errors. 

I look forward to reading the next version of this paper!

Round 2

Reviewer 2 Report

Comments and Suggestions for Authors

Thank you for your efforts revising this paper. All of my comments have been addressed and I now deem it suitable for publication.